# Antibacterial and Antibiofilm Activity of *Ficus carica*-Mediated Calcium Oxide (CaONPs) Phyto-Nanoparticles

**DOI:** 10.3390/molecules28145553

**Published:** 2023-07-20

**Authors:** Asif Ullah Khan, Tahir Hussain, Mubarak Ali Khan, Mervt M. Almostafa, Nancy S. Younis, Galal Yahya

**Affiliations:** 1Department of Microbiology, Abdul Wali Khan University, Mardan 23200, Pakistan; asifukm486@gmail.com; 2Department of Physical Chemistry and Technology of Polymers, Silesian University of Technology, 44-100 Gliwice, Poland; 3Joint Doctoral School, Silesian University of Technology, Akademicka 2A, 44-100 Gliwice, Poland; 4Department of Biotechnology, Abdul Wali Khan University, Mardan 23200, Pakistan; makhan@awkum.edu.pk; 5Department of Chemistry, College of Science, King Faisal University, Al-Ahsa 31982, Saudi Arabia; malmostafa@kfu.edu.sa; 6Department of Pharmaceutical Sciences, College of Clinical Pharmacy, King Faisal University, Al-Ahsa 31982, Saudi Arabia; nyounis@kfu.edu.sa; 7Department of Microbiology and Immunology, Faculty of Pharmacy, Zagazig University, Al Sharqia 44519, Egypt; galalyehia@zu.edu.eg

**Keywords:** nanoparticles, characterization, biomedical applications, antimicrobial, antibiofilm

## Abstract

The significance of nanomaterials in biomedicines served as the inspiration for the design of this study. In this particular investigation, we carried out the biosynthesis of calcium oxide nanoparticles (CaONPs) by employing a green-chemistry strategy and making use of an extract of *Ficus carica* (an edible fruit) as a capping and reducing agent. There is a dire need for new antimicrobial agents due to the alarming rise in antibiotic resistance. Nanoparticles’ diverse antibacterial properties suggest that they might be standard alternatives to antimicrobial drugs in the future. We describe herein the use of a *Ficus carica* extract as a capping and reducing agent in the phyto-mediated synthesis of CaONPs for the evaluation of their antimicrobial properties. The phyto-mediated synthesis of NPs is considered a reliable approach due to its high yield, stability, non-toxicity, cost-effectiveness and eco-friendliness. The CaONPs were physiochemically characterized by UV-visible spectroscopy, energy-dispersive X-ray (EDX), scanning-electron microscopy (SEM), X-ray diffraction (XRD), and Fourier-transform infrared spectroscopy (FTIR). The biological synthesis of the calcium oxide nanoparticles revealed a characteristic surface plasmon resonance peak (SPR) at 360 nm in UV-Vis spectroscopy, which clearly revealed the successful reduction of the Ca^2+^ ions to Ca^0^ nanoparticles. The characteristic FTIR peak seen at 767 cm^−1^ corresponded to Ca-O bond stretching and, thus, confirmed the biosynthesis of the CaONPs, while the scanning-electron micrographs revealed near-CaO aggregates with an average diameter of 84.87 ± 2.0 nm. The antibacterial and anti-biofilm analysis of the CaONPs showed inhibition of bacteria in the following order: *P. aeruginosa* (28 ± 1.0) > *S. aureus* (23 ± 0.3) > *K. pneumoniae* (18 ± 0.9) > *P. vulgaris* (13 ± 1.6) > *E. coli* (11 ± 0.5) mm. The CaONPs were shown to considerably inhibit biofilm formation, providing strong evidence for their major antibacterial activity. It is concluded that this straightforward environmentally friendly method is capable of synthesizing stable and effective CaONPs. The therapeutic value of CaONPs is indicated by their potential as a antibacterial and antibiofilm agents in future medications.

## 1. Introduction

Nanotechnology is the study at atomic and subatomic levels of nanosized particles ranging from 1–100 nm in size, called nanoparticles, which have a wide range of applications in biomedical fields [1]. Metallic nanoparticles are useful in medication delivery, biologically tailored treatments, cosmetics, and electronics [2,3,4,5]. Furthermore, NPs can be synthesized by physical and chemical methods, but these approaches are expensive and non-eco-friendly. The synthesis of NPs from plant extracts is preferable to chemical synthesis due to the high yield, stability, non-toxicity, cost efficiency, and eco-friendliness [6,7,8,9,10,11]. The green approach is also chosen since it does not need maximal temperatures, pressure, or hazardous chemicals, which are needed in chemical and physical methods of NP production [12]. Plant extracts are utilized as cappers and reducers in green-chemistry processes to biologically prepare nanoparticles, such as titanium dioxide (TiO_2_), silver (Ag), iron (Fe), tin (Sb), calcium (Ca), copper oxide (CuO), nickel oxide (NiO), and zinc oxide (ZnO) [7].

Nanoparticles have innate cytotoxic properties, which contribute to their anticancer and antibacterial efficacy against cancer cells and other microbes, including bacteria and fungi [13]. Moreover, CaONPs have already been biosynthesized from extracts from various plants, they can be used as capping and reducing agents, and their antimicrobial potential has been recognized for decades in the medical field, from both traditional and scientific perspectives [6]. This suggests that they may be a useful and convenient remedy or choice for treating multidrug-resistant (MDR) bacterial isolates [14]. 

The fig tree, or fig for short, is the popular name for the flowering plant, *Ficus carica*, which is indigenous to the Mediterranean and Western Asian regions. *Ficus carica* has been studied for its wide range of uses due to its well-documented medicinal properties. For centuries, *Ficus carica’s* medicinal importance has been recognized, and the plant has been used to treat a wide range of diseases, including those related to the cardiovascular and respiratory systems, the digestive tract, and the nervous system [15,16,17]. *Ficus carica’s* antibacterial, antiviral, antioxidant, and antifungal properties are due to the presence of phytochemicals, such as flavonoids, alkaloids, and phenolic acids [18,19].

In designing the present study, we kept in mind the novel and consequential function of NPs in the biological and therapeutic arenas. *Ficus carica* extracts were used as stabilizing and reducing agents to biosynthesize CaONPs; following their biological processing, CaONPs were analyzed for characterization using state-of-the-art methods, such as ultraviolet-visible spectroscopy, energy-dispersive X-ray analysis, scanning-electron microscopy, X-ray diffraction, and Fourier-transform infrared spectroscopy. Finally, the antibacterial and antibiofilm potential of the CaONPs was tested against a number of different bacterial isolates.

## 2. Results and Discussion

### 2.1. Plant-Extract Preparation and Biosynthesis of Calcium Oxide NPs

As shown in Figure 1, the preparation of the *Ficus carica* extract was carried out under the most favorable conditions, and the pH of the solution was maintained at 5.7 throughout. This was because an acidic pH makes the process of bio-component denaturation significantly simpler. The extracts of *F. carica* are rich in phytochemical components, including flavonoids and phenolic acids, which are the most important contributing chemicals that take part in the manufacturing of NPs [20,21]. All the bio components that have been discussed up to this point are very effective as capping and reducing agents of metal ions, and as a result, they are directly involved in the biological formation of nanomaterials [12]. Because the biological production of nanoparticles is time-dependent, it was observed that a higher rate of reduction took place after a reaction had taken place for a period of two hours. The plant biomolecules successfully reduced the calcium to calcium oxide nanoparticles, as evidenced by the color change of the mixture to dark brown [8]. This was an indication of the calcium ions’ reduction to CaONPs. Further, the confirmation of the CaONPs was performed using UV-spectroscopy. 

### 2.2. UV-Visible Spectroscopic Analysis

The UV-visible spectroscopy is a technique in which the beams of rays are absorbed by a solution in characteristic ranges comprising, both visible and UV radiation while generating certain specific peaks [22]. This radiation is essentially responsible for providing imminent activation energy, resulting in electron excitation in various atomic orbits. This modern form of analysis is mainly involved in recognizing or identifying chemical compounds as well as analyzing samples quantitatively [23]. For the identification of sample concentrations, the values of UV and visible light absorbance and their specific peaks formed by the above-mentioned spectral ranges are identified via the Beer–Lamberts law [23]. A dual-beam spectrophotometer (UV-2450) was used in our experimental work in order to measure the absorbance vs the wavelength curve of the biosynthesized NPs. The reduction of the Ca^2+^ ions was observed through the measurement of the UV-visible spectrum. In the environmentally friendly reaction between the *F. carica* extract and the solution of calcium nitrate tetrahydrate, the reduction of the Ca^2+^ ions to Ca^0^ was screened with the help of the UV-vis spectrum in a standard and characteristic spectral range of 200 nm to 800 nm, as in [24]. 

Figure 2a shows that the biological synthesis of the calcium oxide nanoparticles possessed a characteristic surface plasmon resonance peak (SPR) at 360 nm, which is similar to the results in [25], clearly revealing the successful reduction of the Ca^2+^ ions to Ca^0^ nanoparticles due to the biocomponents present in the *F. carica* extracts, in line with [26]. This confirmed the CaONPs’ biosynthesis. 

### 2.3. Fourier-Transform Infrared Spectroscopy (FTIR) Analysis

The FTIR spectroscopy was conducted in the spectral range from 400 cm^−1^ to 4000 cm^−1^. Its purpose was to determine the possibility of the involvement of various functional groups of plant extract responsible for the reduction process of metallic ions and the biosynthesis of CaO nanoparticles. Major peaks appeared at 661 cm^−1^, showing stretching-vibration peaks corresponding to Ca-Halogen, 767 cm^−1^ (Ca-O-bond stretching)**,** 809 cm^−1^, 897 cm^−1^ (stretching-vibration peaks corresponding to C-H-, C-C-, and Ca-O-Ca bonding), 954 cm^−1^ (representing the OH of alcohol), 1195 cm^−1^ (for the amine functional group), 1503 cm^−1^, 2574 cm^−1^ (peaks corresponding to COOH), 2945 cm^−1^ (C-H stretching), 3386 cm^−1^ (the N-H-, C-H-, and O-H-bond stretching of amines and amides), and 3696 cm^−1^ (amide), respectively, as shown in Figure 2b, which shows similarities with [24]. The characteristic peak seen at 767 cm^−1^ showed Ca-O bond stretching, which confirmed the biosynthesis of the calcium oxide nanoparticles. Extracts from *F. carica* and other plants contain various bio components or phyto-constituents, like flavonoids and phenolic acids, as well as some other functional groups present in these polymers and certain protein-based matter, etc., which are involved in the reduction of metallic ions and NP biosynthesis [27].

### 2.4. X-ray Powder Diffraction (XRD) Analysis

This is a rapid analytical approach, which is very helpful in determining the unit cells and crystalline nature of samples. While performing the XRD, the sample was converted into powdered form and homogenized in order to identify the bulky composition of the biologically prepared CaONPs. The crystalline nature of the CaONPs was determined via the XRD pattern, which clearly revealed that the CaONPs were actually crystalline in nature. Various diffraction planes were seen at (111), (200), (222), (202), and (311) [28,29,30,31], as shown in Figure 2c. The average size of the biosynthesized CaO nanoparticles, which was 68.6 nm, was determined using the Scherrer equation. It was observed that the small NPs possess more efficacy in biomedical applications, but it should be remembered that the size of nanomaterials is dependent on the metallic-ion reduction rate by plant extracts [32,33].

### 2.5. Energy-Dispersive X-ray Spectroscopy (EDX) Analysis

In order to perform this analysis, the NPs were bombarded by beams of electrons. As a result, X-rays were emitted from the sample; these were traced through a detector and utilized for the production of an output curve, which provided information regarding the elemental composition of the biosynthesized nanomaterials. The SEM was performed in combination with an EDX analysis. The output spectra clearly revealed the prominent peaks of the CaONPs, as shown in the Figure 2d. Many other peaks were also observed that showed the presence of calcium, oxygen, sodium, carbon, magnesium, phosphorus, and sulfur. The additional EDX peaks were obtained through the plant extract, and demonstrated a major contribution with the bio-components, participating in the biosynthesis of the CaONPs. The highly intensive peaks of the calcium oxide confirmed the biological preparation of the CaONPs, which showed a close agreement with the results in [27,32]. The atomic weight percentages of the calcium and oxygen were 31.40% and 58.68%, respectively.

### 2.6. Scanning-Electron Microscopy (SEM) Analysis

The SEM analysis was performed to determine the physical dimensions and morphological characteristics of the *F. carica*-mediated CaONPs. The white patches seen in the electronic micrographs and shown in Figure 3 essentially indicated the biosynthesis of the CaONPs, using the plant extract. The analysis of the SEM images was performed with Image J v1.54f, NIH, US which helped to determine the average particle diameters of the CaONPs. As shown in Figure 3, some irregularly shaped and non-uniformed particles were observed, with an average diameter of 84.78 ± 2.0 nm. While utilizing Image J software, 50 particles from every SEM micrograph were analyzed, and their mean was taken as the average diameter of the CaONPs. The bright areas in the SEM images show high emissions of secondary electrons upon their exposure to the electron beam. All this occurred because of the increased surface-area-to-volume ratio in these patches [31]. The SEM micrographs indicated the agglomeration of the CaONPs, which were due to the interactions between these NPs, which were similar to the findings in [34,35,36]. The biosynthesized nanomaterials comprised irregular rounded grains, as shown below, in magnification. 

### 2.7. Antibacterial Activities of CaONPs

To test the antibacterial potential of the CaONPs, an agar-well-diffusion assay was used. While performing this assay, inhibition zones of 28 ± 1.0, 23 ± 0.3, 18 ± 0.9, 13 ± 1.6, and 11 ± 0.5 mm were observed against *P. aeruginosa, S. aureus, K. pneumoniae, P. vulgaris*, and *E. coli*, respectively, as shown in the Figure 4. Furthermore, the de-ionized water did not show any antibacterial activities against these bacterial isolates; hence, it was used as a negative control, while ciprofloxacin was used as a positive control in the experiment, showing 25 ± 3.0 ZI against all the selected bacteria. The amount of calcium oxide NPs applied the during antibacterial analysis was the same for all the bacterial spps. The antibacterial potential of the nanomaterials was mainly attributed to their interaction and damage to the plasma membrane, resulting in the oozing of the intracellular contents out of the microbial organisms [37]. Researchers reported the antibacterial properties of CaONPs against Gram-positive and Gram-negative bacteria in a dose-dependent manner [18,38,39].

Factors such as the nanoparticle size, concentration, surface characteristics, and targeted bacterial strain may affect the antibacterial efficacy of calcium oxide nanoparticles. There is a need for further studies of the cytotoxicity and biocompatibility of calcium oxide nanoparticles before they can be used optimally in antibacterial applications. The CaONPs showed antibacterial activity either through interactions with the cell membrane of the bacteria, which caused the membrane to become disrupted and the contents of the cell to leak out, or by releasing calcium ions (Ca^2+^) and hydroxide ions (OH), increasing the pH that resulted from the hydroxide ions. These increases in pH may cause bacterial cellular processes to become dysfunctional, which in turn threatens bacteria’s ability to survive. Furthermore, CaONPs also have the potential to produce reactive oxygen species. These reactive species have the potential to trigger oxidative stress, which ultimately results in the death of bacterial cells.

### 2.8. Anti-Biofilm Activities of CaO NPs

Biofilms are colonies of bacteria that may grow on a variety of surfaces, including medical devices and implants, which can cause infections that last for extended periods. Calcium oxide nanoparticles have shown the capacity to both prevent the production of new biofilms and disrupt the formation of existing biofilms. As a result, these nanoparticles have the potential to be useful in both the prevention and the treatment of infections caused by biofilms. Biofilms are colonies of microorganisms, such as bacteria, that attach to surfaces and create protective matrices [40,41,42]. These communities may be rather complicated. The elimination of biofilms is notoriously challenging, and their presence may contribute to recurrent infections, as well as antibiotic resistance [42]. Nanoparticles of calcium oxide display antibiofilm action in a number of different ways, some of which are the disruption of biofilm matrix, the production of reactive oxygen species (ROS), interruptions in the formation of cell membranes, changes in pH, direct contact with cells, etc. [43,44,45,46]. In this research study, an in vitro anti-biofilm analysis of CaONPs was performed in a dose-dependent manner against selected microorganisms. The results of this activity clearly showed that the biosynthesized CaONPs inhibited the synthesis of the bacterial biofilm compared to the negative control. The treatment of the *P. aeruginosa* for 3 days with *Ficus carica* mediated the biosynthesized calcium oxide nanomaterials (100 µg/mL) and inhibited biofilm production by >99%. The average MIC (IC_50_) value of the CaONPs required to reduce the biofilm synthesis was 62.5 ± 0.2 µg/mL. Against *S. aureus, K. pneumoniae*, *P. vulgaris*, and *E. coli*, these CaONPs revealed potential antibiofilm activities, as shown in Table 1 and Figure 5. Nanoparticles that target antibiofilm treatment have seen an enormous surge in popularity over the last decade. These particles are reactive entities and have the ability to quickly permeate bacterial matrices. To stop the development of biofilms, biomedical surfaces can also be nano-functionalized via coating, impregnation, or embedding with nanomaterials [43,47,48,49].

The findings of this research project made it evident that biosynthesized calcium oxide nanoparticles are very effective antibacterial agents, and that *Ficus carica* is an excellent reducing and stabilizing agent. These findings support our ecofriendly approach to the synthesis of nanomaterials mediated by phytoconstituents.

## 3. Materials and Methods

### 3.1. Ficuscarica-Extract Preparation

*Ficus carica* fruit was purchased from a local market, and confirmed by the botanist in the Department of Botany, Abdul Wali Khan University Mardan, Pakistan. A total of 50 g of *F. carica* was cut into tiny pieces and ground into paste using a grinder. The paste was gently mixed with 200 mL of de-ionized water and boiled at 60 °C for 30 min [50,51]. The extract was filtered thrice using Whatman filter paper no.1 to remove residual remainders, followed by centrifugation at 5000 rpm for 15 min. Finally, the supernatant was collected in a separate tube and stored at 4 °C. 

### 3.2. Biosynthesis of Calcium Oxide NPs

*Ficus carica* extract was used for the biosynthesis of calcium oxide NPs. In total, 50 mL of 1 mM of calcium nitrate salt (CaNO_3_)_2_·4H_2_O and 50 mL of *F. carica* extract were mixed thoroughly in a flask at (pH: 5.7) and stirred for 2 h at 60 °C [52]. The production of nanoparticles was shown to have occurred when there was a noticeable change in the color of the mixture to dark brown. The mixture was centrifuged for 15 min at a speed of 10,000 revolutions per minute (rpm). The pellets were collected and purified through calcination at 500 °C in order to obtain pure nanoparticles. 

### 3.3. Characterization of Biosynthesized Calcium Oxide NPs

Energy-dispersive X-ray (EDX), UV-visible spectroscopy, Scanning-electron microscopy (SEM), X-ray diffraction (XRD), and Fourier-transform infrared spectroscopy (FTIR) were used for the physio-chemical characterization of calcium oxide nanoparticles. In order to examine the eco-friendly nature of the interaction that took place between plant extract and calcium nitrate salt, the biologically generated CaONPs were subjected to UV-spectroscopy in the frequently utilized spectral range of 200–800 nm [53,54]. For the identification of crystalline phase of CaONPs, XRD analysis was performed using Panalytical’s X’Pert X-ray diffractometer. The crystallized sizes of NPs were calculated by using Debye–Scherer’s equation [19,54]:(1)D=kλ/βCosθ
where *k* is instrument constant (0.9), *λ* is wavelength of *X* ray diffraction (0.1541 nm), ‘*β*’ is FWHM (full width at half maximum), ‘*θ*’ is the diffraction angle, and ‘*D*’ is particle diameter (nm).

In order to find different chemical functional groups vigorously involved in the biological preparation of CaONPs, Fourier-transform infrared-spectroscopy analysis was performed using IRTracer-100 FTIR spectrometer in the spectral range of 400 to 4000 cm^−1^. Physical and morphological features of the biosynthesized NPs, such as shape and diameter, were examined through Scanning-electron microscopy (JSM-5910, JEOL, Tokyo, Japan). CaONPs were exposed to Energy-dispersive X-ray spectroscopy (INCA200, Oxford instruments, Abingdon, UK) for the determination of their elemental composition.

### 3.4. Collection of Bacterial Samples

Pre-identified bacterial species, including *E. coli, P. aeruginosa, S. aureus, K. pneumoniae*, and *P. vulgaris* were collected from Islamabad Diagnostic Center, Islamabad, Pakistan. Fresh culture of these bacterial species was grown on Mueller Hinton agar plates to assess the antimicrobial potential of CaONPs. 

### 3.5. Antibacterial Efficacy of CaO-NPs established by Using Agar-Well-Diffusion Method

The CaONPs’ antibacterial potential was tested against *E. coli, P. aeruginosa, S. aureus, K. pneumoniae*, and *P. vulgaris* and, for this purpose, agar-well-diffusion method was used [55]. Stock bacterial suspension of 0.5 OD McFarland was prepared and 100 µL was poured into Mueller Hinton agar plate via pipette; lawn preparation was conducted by spread-plate technique according to [56]. Next, using a sterile well borer, 5–6 mm wells were made in plate and 100 µL of CaONPs were supplied into each well, and labeled accordingly. In addition, the antibacterial test was conducted with ciprofloxacin serving as the positive control and distilled water serving as the negative control. The plates were allowed to remain in an incubator at 37 °C for 24 h. After incubation, zones of inhibition were measured in mm, according to [57,58,59]

The CaONPs’ antibiofilm potential was tested against *E. coli, P. aeruginosa, S. aureus, K. pneumoniae*, and *P. vulgaris*. The activity was performed in a 96-well microtiter plate, according to [60,61]. The wells were supplied with 100 µL of 0.5 OD bacterial suspension and 60 µL of CaONPs followed by incubation at 37 °C for 24–72 h. After incubation, the crystal-violet assay was used to quantify the biofilm. The wells were washed with deionized water, followed by incubation in methanol for 15 min. The wells were then stained by 100 µL of 1% w/v crystal violet for half an hour at room temperature. The stain was then dissolved in acetic acid (33%) and the readings were taken at 570 nm using Bio Rad microplate reader. The calculation in percentage of bacterial biofilm inhibition was performed using the following formula [39,62].
(2)$$\%\;\mathrm{of}\;\mathrm{Inhibition}=\frac{OD_{Negative\;Control}-OD_{Experimental}}{OD_{Negative\;Control}}\times 100$$

### 3.6. Statistical Analysis

The statistical analysis of our findings was carried out with the assistance of a variety of software programs, including Microsoft Excel, ORIGIN, and IMAGE-J SOFTWARE. A significance level of *p* less than 0.05 was considered to be the normative value.

## 4. Conclusions

While performing this study, we reached the conclusion that the biosynthesis of calcium oxide nanoparticles from *Ficus carica* extracts is a research area of interest. Fig-plant-mediated biosynthesized CaONPs play a prominent role in biomedical applications, as they are efficient therapeutic agents. Our results show that CaONPs have immense antibacterial and anti-biofilm potential against many bacterial isolates, while CaONPs are recommended to as potent therapeutic agents for use in nanomedicines and can also be utilized as nano-functionalized coatings in medical implants. However, since we performed all the biomedical analyses and applications of the biosynthesized NPs in vitro, it is strongly recommended that proper in vivo tests are performed to investigate the biomedical applications of CaONPs. While calcium oxide nanoparticles have the potential to produce a number of positive outcomes, more studies are required to fully understand their characteristics, behavior, and hazards in a variety of contexts. The safe and responsible usage of nanoparticles necessitates the consideration of appropriate safety measures and risk evaluations. 

## Figures and Tables

**Figure 1 molecules-28-05553-f001:**
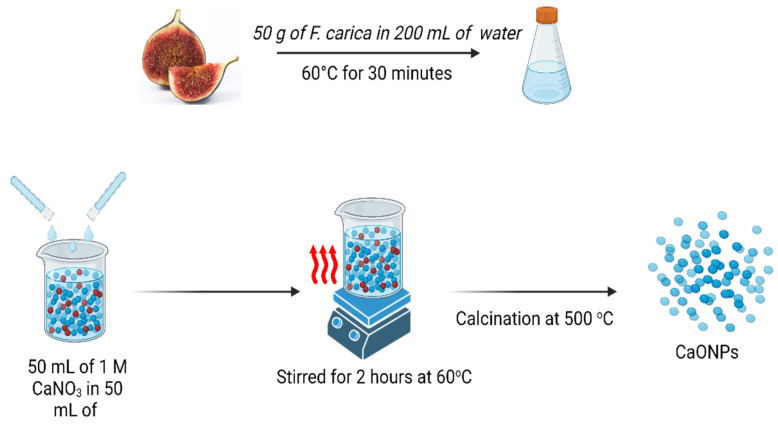
Schematic diagram of *F. carica*-extract preparation and calcium-oxide-nanoparticle synthesis. Created with BioRender.com.

**Figure 2 molecules-28-05553-f002:**
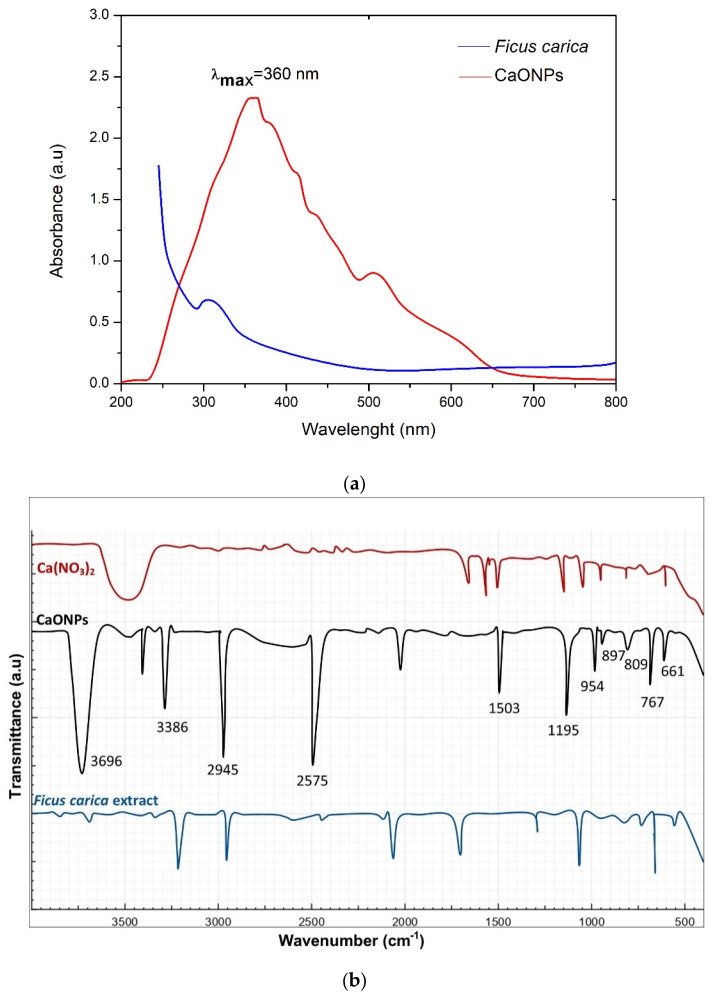
Different characterization analyses of CaONPs. (**a**) UV-Vis spectroscopy, (**b**) FTIR, (**c**) X-ray diffraction, and (**d**) EDX analysis of CaONPs.

**Figure 3 molecules-28-05553-f003:**
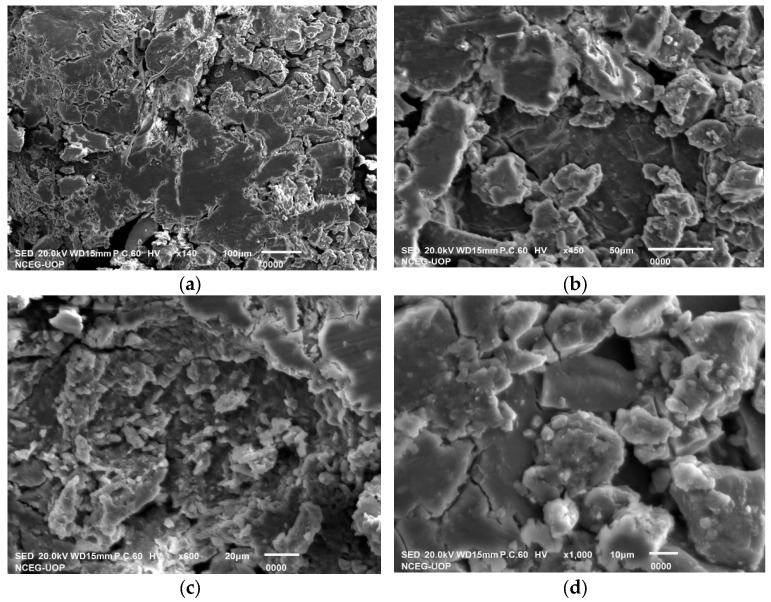
SEM micrographs for CaONPs at different magnifications; (**a**) 140×, (**b**) 450×, (**c**) 600×, and (**d**) 1000×.

**Figure 4 molecules-28-05553-f004:**
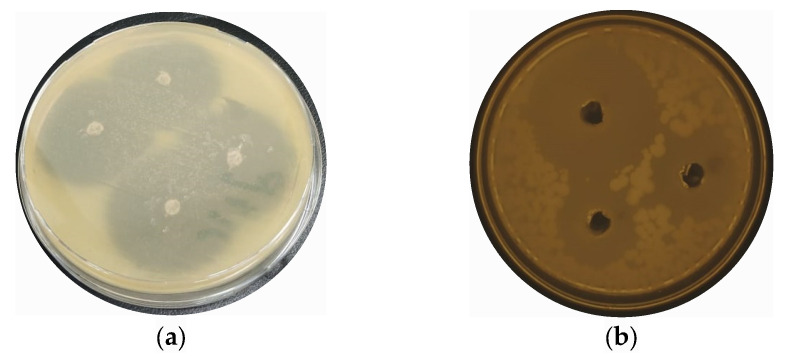
Antibacterial potential of biosynthesized CaONPs of *P. aeruginosa* (**a**), *S. aureus* (**b**), *K. pneumoniae* (**c**), *P. vulgaris* (**d**), and *E. coli* (**e**). (**f**) The graphical representation of antibacterial activity of CaONPs.

**Figure 5 molecules-28-05553-f005:**
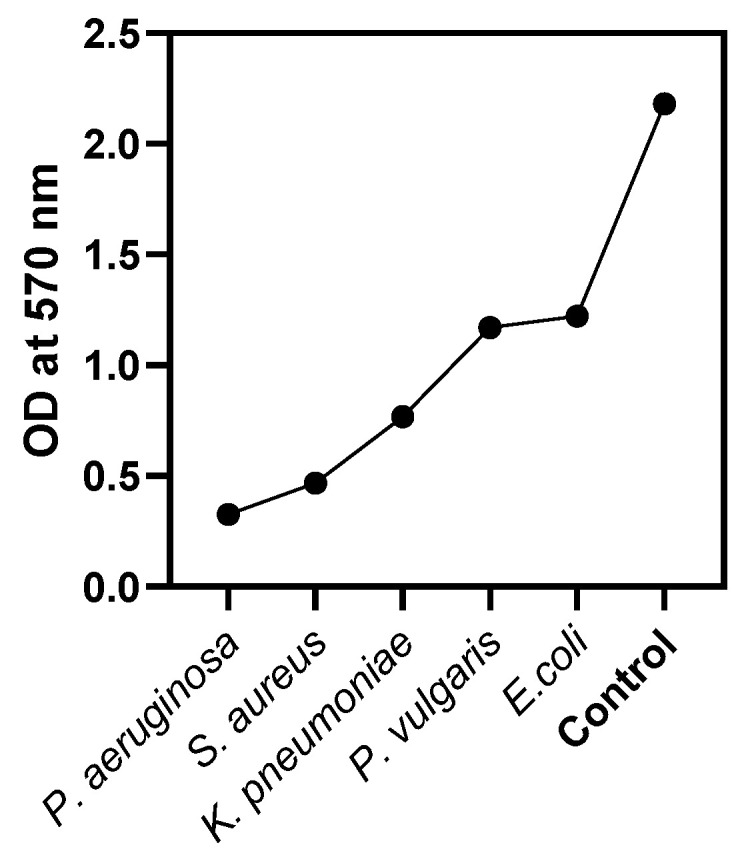
Antibiofilm assay, absorbance at 570 nm by selected microorganisms.

**Table 1 molecules-28-05553-t001:** Bio-film-formation inhibition by CaONPs.

Bacteria	OD at 570 nm	Percentage Inhibition of Biofilm by CaONPs
** *P. aeruginosa* **	0.326	99 ± 1.4
** *S. aureus* **	0.468	98 ± 0.3
** *K. pneumoniae* **	0.767	79 ± 0.9
** *P. vulgaris* **	1.169	67 ± 0.9
** *E. coli* **	1.221	61 ± 1.6
**Growth Control**	2.181	

## Data Availability

All the data are available within the manuscript. Additional data will be provided upon request from the corresponding author.

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
