# Peer review of "Antibacterial and Antibiofilm Activity of Ficus carica-Mediated Calcium Oxide (CaONPs) Phyto-Nanoparticles"

_molecules, 2023, doi:10.3390/molecules28145553_

Round 1

Reviewer 1 Report

This study was inspired by the significant role of nanomaterials in biomedicines. The researchers successfully synthesized calcium oxide nanoparticles (CaONPs) using a green chemistry approach and Ficus carica extract as a capping and reducing agent. The CaONPs exhibited strong antibacterial properties, inhibiting the growth of various bacteria, and effectively preventing biofilm formation, indicating their potential as alternative antimicrobial agents in future medicine. The study highlights the importance of environmentally friendly methods for synthesizing stable and effective nanomaterials with therapeutic applications. The topic falls within the scope of the Molecules journal. That being said, my enthusiasm for endorsing the publication of the manuscript in its present form is tempered by several issues and concerns that must be adequately addressed by the authors as detailed below.

1.    Provide an explanation of why 72 hours was selected as an incubation time for the bacteria as opposed to a more conventional 18-24hr endpoint utilized for multi-well plate testing methods. To which stage of the bacteria lifecycle does this time correspond?

2.    Characterize the EDX results by the percentage of each element present.

3.    Label the corresponding characteristic peaks in the FTIR spectra (Figure 2b).

4.    It is hard to see the scale in Figure 3. Please add the appropriate label to the bar.

5.    Provide an explanation for the observed biofilm reduction of different bacterial species. Are the trends similar for gram-negative bacteria? Are they similar to gram-positive bacteria or not?

Minor revision of English is required.

Author Response

  1. the cultures were incubated for three days (72 hours) in order to justify the formation of biofilm. On day three the highest biofilm formation was observed as compared to 1st and 2nd It corresponds to the stationary phase of the bacteria lifecycle as in this stage the bacterial biofilm formation is on the peak.
  2. EDX results were characterized by percentage in the revised version of the manuscript
  3. FTIR peaks have been labeled
  4. High-resolution pictures have been provided in the revised version of the manuscript
  5. the formation of biofilm by various bacteria was inhibited vigorously and both gram positive as well as gram negative bacterial biofilms production was reduced, hence they are effective against both types.
  6.  

Reviewer 2 Report

I do not recommend the current version of the manuscript. The following points are my comments.

1.      The typesetting, spelling, and grammatical English expression of this manuscript need improvement.

2.      In the abstract and other sections, the abbreviation should be defined only once when it appears for the first time and consistently used later throughout the manuscript (e.g. CaONPs, nanoparticles, all bacteria names, F. carica, and all characterization techniques).

3.      The abstract must include only significant results but not in detail.

4.      The use of words should be consistent: NPs / nanoparticles, CaO-NPs / CaO NPs / calcium oxide NPs, calcium nanoparticles / calcium NPs / calcium oxide nanomaterials.

5.      Citing reference no. 22 seems unreasonable.

6.      Equation 1 uses k but the definition is K. Also, the unit of D must be specified.

7.      Section 3.1, “All of the bio components that have been discussed up to this point”, only F. carica was mentioned.

8.      Figure 1, what are the blue and red spheres in the beaker?

9.      Section 3.2 page 5, “Reduction of Ca⁺2 ions was observed through measurement of UV”, only 1 spectrum was shown with one lamda max. What did the authors mean about the reduction? The UV spectra from several time intervals must be shown.

10.  Correct Ca+2 to Ca2+.

11.  Section 3.5, SEM EDX only confirmed the existence of Ca and O. In order to prove that the product was CaO, XPS must be tested.

12.  Figure 2b, why were the baselines of FTIR spectrum not at the same level?

13.  Section 3.6 page 6, “Some irregularly shaped and non-uniformed particles observed with an average diameter of 84.78±2.0 nm.”, this statement was not supported by the provided SEM showing only agglomeration. High-resolution SEM and TEM are required to prove that the size of NPs is in the range of nanoscale.

14.  Figure 3, the size of the scale bar is not clear, the magnification must be specified in the caption. Figures D and E are from the same area, what is the purpose of this?

15.  Figure 5, the display of the petri dish looks like it was squeezed.

16.  In conclusion, the author mentions Fitus carica plant and then fig plant. Are they the same plant?

English grammar, spelling, and typesetting need improvement throughout the manuscript.

Author Response

Response to reviewers' comments

  1. The typesetting, spelling, and grammatical English expressions were rectified and improved.
  2. In the abstract and other sections, the abbreviations are now defined only once when it appears for the first time and consistently not repeated again.
  3. In the abstract, only significant results were included.
  4. The use of words was made consistent: NPs / nanoparticles, CaO-NPs / CaO NPs / calcium oxide NPs, calcium nanoparticles/calcium NPs / calcium oxide nanomaterials.
  5. Citing reference no. 22 was correlated
  6. In equation 1 K was corrected. Also, the unit of D is in nanometres.
  7. Section 3.1, “All of the bio components means those present already in F. carica plant extract as were mentioned in the previous sentence.
  8. Figure 1, the blue and red spheres in the beaker show calcium nitrate salt and F. carica extracts respectively, it is just an illustration.
  9. Dear reviewer, we just included the spectra upon which nanoparticle production occurs, however, we have revised the figure.
  10. corrected
  11. The figure was updated and revised
  12. The figure has been updated 
  13. high-resolution images have been provided in the revised version of the manuscript with clear scale bars from which the size can be calculated
  14. The figures have been updated
  15. the correct form of images has been provided
  16. Yes, they are the same plant

Reviewer 3 Report

Manuscript details: Journal: Molecules
Manuscript ID: molecules-2481244; Type of manuscript: Article;

Topic: Antibacterial and Antibiofilm Activity of Edible Fruit Ficus carica-Mediated Calcium Oxide (CaONPs) Phyto-Nanoparticles

The manuscript entitled “Antibacterial and Antibiofilm Activity of Edible Fruit Ficus carica-Mediated Calcium Oxide (CaONPs) Phyto-Nanoparticles” is quite significant in the field of Nanotechnology.  Calcium oxide nanoparticles (CaONPs) have been biosynthesized and characterized effectively. Calcium oxide nanoparticles (CaONPs) were effective against bacteria. The antibacterial and anti-biofilm analysis showed inhibition of bacteria in mm: P. aeruginosa (28±1.0)> S. aureus (23±0.3) > K. pneumoniae (18±0.9)> P. vulgaris (13±1.6)> E. coli (11±0.5) mm.

The manuscript can be accepted for publication after major modifications in the light of the following comments and feedback. The authors are advised to submit the revised manuscript for final acceptance.

1.      Abstract: UV-vis peak can be included.

2.      2.1. Ficus carica………………….. line 4, include the said reference as reference No. 22 (https://www.mdpi.com/1424-8247/15/6/760) for sample preparation and extraction purposes.

3.      2.2. Biosynthesis…………………3rd line, correctly write pH

4.      2.3. Characterization…………………. Consider the said reference as reference No. 23 (https://link.springer.com/article/10.1007/s10661-021-09301-w) for crystallite size calculation of NP.

5.      2.6. Anti-biofilm………….line 5: correct followed

6.      3.4. X-ray powder…………….line 11, include the already said reference (no. 23) (https://link.springer.com/article/10.1007/s10661-021-09301-w) too.

7.      Fig. 2 (a) X-axis, correctly write wavelength; If possible, include UV-vis spectra of plant extract and Ca(NO3)2 in the same figure too.

8.      Fig. 2 (b) If possible, include IR spectra of plant extract and Ca(NO3)2 too in the same figure.

9.      Fig. 2 (c) In the discussion part, include all 2θ values.

10.  Fig. 2 (d) If possible, include original graph along with % composition details of involving components. Include EDX spectra of plant material too for comparison purpose.

11.  Fig. 3. Unable to see any value of magnification and SEM images in 100 nm size range of CaONP. Keep here 1 magnified and clear SEM image of plant material and CaO NP.

12.  3.7. Antibacterial…………………..Include comparative data table of zone of inhibition in mm for different taken bacteria. It is important to compare the zone of inhibition of CaO NP with control (antibiotic), plant extracts, and calcium nitrate too.

Author Response

  1. Abstract: UV-visible spectroscopy peak was included.
  2. 2.1. Ficuscarica…………………. line 4, included the said reference for sample preparation and extraction purposes.
  3. 2.2. Biosynthesis…………………3rdline, pH was written correctly.
  4. 2.3. Characterization………………. The said reference was considered as  for the crystallite size calculation of the NP
  5. 2.6. Anti-biofilm…………. line 5: corrected the spelling followed.
  6. 3.4. X-ray powder……………. line 11, included the already said reference as well.
  7. The figures have been updated
  8. FTIR peaks of CaONPs, CaNO3, and Plant extract have been provided in the revised version of the manuscript
  9. added
  10. An elemental percentage of the elements has been provided in the revised version of the manuscript
  11. High-resolution SEM images have been provided in the revised version of the manuscript
  12. the results have been explained with the reference to the inhibition zones of control

Round 2

Reviewer 2 Report

In the final proof process, complete the reference details, e.g. page number, of all references.

Reviewer 3 Report

Greetings!

Accepted for publication.